# The impact of perinatal maternal stress on the maternal and infant gut and human milk microbiomes: A scoping review protocol

**Niamh Ryan** [1]*, **Patricia Leahy-Warren**[1], **Helen Mulcahy**[1], **Siobhain O'Mahony**[2], **Lloyd Philpott**[1]

1 School of Nursing and Midwifery, University College Cork, Wilton, Cork, Ireland, 2 Department of Anatomy and Neuroscience, APC Microbiome Ireland, University College Cork, Ireland

* 122117697@umail.ucc.ie

## Abstract

### Objective

The objective of this scoping review is to review the research evidence regarding the impact of perinatal maternal stress on the maternal and infant gut and human milk microbiomes.

### Introduction

Perinatal stress which refers to psychological stress experienced by individuals during pregnancy and the postpartum period is emerging as a public health concern. Early exposure of infants to perinatal maternal stress can potentially lead to metabolic, immune, and neurobehavioral disorders that extend into adulthood. The role of the gut and human milk microbiome in the microbiome-gut-brain axis as a mechanism of stress transfer has been previously reported. A transfer of colonised aberrant microbiota from mother to infant is proposed to predispose the infant to a pro-inflammatory microbiome with dysregulated metabolic process thereby initiating early risk of chronic diseases. The interplay of perinatal maternal stress and its relationship to the maternal and infant gut and human milk microbiome requires further systematic examination in the literature.

### Inclusion criteria

This scoping review is an exploratory mapping review which will focus on the population of mothers and infants with the exploration of the key concepts of maternal stress and its impact on the maternal and infant gut and human milk microbiome in the context of the perinatal period. It will focus on the pregnancy and the post-natal period up to 6 months with infants who are exclusively breastfed.

### Methods

This study will be guided by the Joanna Briggs Institute's (JBI) methodology for scoping reviews along with use of the Prisma Scr reporting guideline. A comprehensive search will be conducted using the following databases, CINAHL Complete; MEDLINE; PsycINFO,

**Data Availability Statement:** All data are in the manuscript and supporting information files.

**Funding:** The author(s) received no specific funding for this work.

**Competing interests:** The authors have declared that no competing interests exist.

Web of Science and Scopus. A search strategy with pre-defined inclusion and exclusion criteria will be used to retrieve peer reviewed data published in English from 2014 to present. Screening will involve a three-step process with screening tool checklists. Results will be presented in tabular and narrative summaries, covering thematic concepts and their relationships. This protocol is registered with Open Science Framework DOI 10.17605/OSF.IO/5SRMV.

## Introduction

Maternal mental health in the perinatal period refers to the psychological well-being of women during pregnancy and the postnatal period [1]. It includes conditions such as depression, anxiety, and stress. Perinatal maternal stress in particular is a public health concern as studies worldwide are reporting a rising prevalence of between 5% - 93% [2–5]. Its rise reportedly influenced by societal pressures, social isolation, economic challenges, pregnancy, and birthing complications [6, 7].

Maternal mental health not only impacts the health of the mother, but it can also have long lasting implications for infant and child development [8–12]. In the past 30 years, a substantial body of evidence is building on Barker's theory of the developmental origin of health and disease (DoHD) which demonstrated that adverse maternal conditions, such as stress during the first 1000 days of life can increase the risk of metabolic, immunologic, and neurobehavioral disorders in offspring [13–22]. Given that perinatal maternal stress is a prevalent and modifiable factor that affects maternal, foetal and child health it requires focused attention in research and preventative health care.

There is growing researcher interest in how stress transfers from the mother to the infant and the role that the gut microbiome plays in the process [23, 24]. The gut microbiome, composed of trillions of microorganisms in the digestive tract, influences gut, immune and metabolic health [25, 26]. The balance of the gut microbiome is crucial for overall well-being, with disruptions linked to various health problems such as cardiometabolic diseases, digestive disorders, neurological disorders [25, 27–33]. There is now evidence that both the gut and human milk microbiome is altered by many maternal factors such as maternal health, maternal body mass index, mode of delivery, and antibiotic use [34–38]. This alteration or dysbiosis in the microbiome caused by aberrant microbiota is proposed to be transferred to the infant, predisposing them to a pro-inflammatory microbiome with dysregulated metabolic processes leading to adverse health conditions. Focusing on the influence of maternal stress, this review is interested in the brain gut axis (the two-way communication between the central nervous system and the gut microbiome and its impact on altering maternal and infant gut and human milk microbiomes. Limited research has focused on how maternal perinatal stress as a single variable affects the microbiomes, with no comprehensive review of existing evidence. Therefore, conducting a scoping review is timely and warranted to systematically map this research area. Seminal work by Arksey and O'Malley describes a scoping review as beneficial for examining the extent, range, and nature of research, while also identifying gaps in the evidence base [39]. This review also highlights the importance of perinatal stress for maternal and infant well-being. It emphasizes the necessity for clinician intervention and highlights research gaps. By tackling perinatal stress, immediate risks such as low birth weight and preterm delivery could be reduced, while also fostering stronger mother-child bonding [40–42]. Furthermore, addressing maternal stress may mitigate negative behaviours like smoking and alcohol

consumption thus further improving lifestyle [43, 44].Given the relative novelty of data in this topic, understanding the depth of evidence through mapping is crucial, as feasibility of a systematic review may be difficult due to a potential lack of research [45]. It is advised that a protocol is essential prior to a scoping review [46].

## Review question

The review question was formulated using the Participant (P), Concept (C), Context (C) framework following JBI guidance [47] (Table 1). The review aims to provide an overview of the evidence to address the following question: What is the impact of perinatal maternal stress on the maternal and infant gut and human milk microbiomes? With the objective to; a) Identify the methods and tools used by researchers to measure stress and the microbiome(s), b) Identity if perinatal maternal stress has an impact on bacterial taxa in the microbiome(s), c) Identify if perinatal maternal stress has an impact on bacterial diversity of the microbiome(s).

## Methods

The scoping review will be conducted following the guidelines for scoping reviews described by the JBI Manual for Evidence Synthesis and Preferred Reporting Items for Systematic Reviews and Meta-analyses extension for Scoping review (PRISMA-ScR) [46, 47].

### Study eligibility criteria

The inclusion exclusion criteria for this review will be based on the population-concept-context framework. (Fig 1) (Table 1) recommended by the Johanna Briggs Institute (JBI) [47].

Agreement among five researchers regarding inclusion and exclusion criteria at the start of the review process occurred, with each researcher agreeing with the final criteria. The eligibility criteria for this study are outlined in Table 2 where it is matched to population concepts and context.

### Types of sources

This review will include primary peer reviewed papers that explored maternal stress, the human milk microbiome, and maternal and infant gut microbiome. Secondary analyses, such as systematic reviews will be included if they meet the inclusion criteria. Case reports, book chapters, guidelines, commentaries, editorials, letters to editors, and narrative reviews are excluded as these sources do not directly address the review question and pose challenges in data extraction. This review will not include grey literature due to the difficulty in retrieving

**Table 1. PCC Framework.**

|  |  | **Evidence to Support PCC** |
|---|---|---|
| **Population** | Mother and infants | Maternal factors are influencing a transmission of risk of chronic diseases to the infant in adverse conditions such as stress via the microbiome [7, 43–46]. |
| **Concepts** | Maternal stress | Exposure to the adverse environment of maternal stress is proposed to alter the maternal microbiome(s), which is implicated in risk transfer to the infant [15, 25, 48, 49]. |
|  | Human milk microbiome |  |
|  | Maternal/infant gut microbiome |  |
| **Context** | Perinatal period | The perinatal period is seen as the time for foundation of health in the first 1000 days, where neural, immune, and metabolic function are laying foundations for infant health [13, 50–52]. |

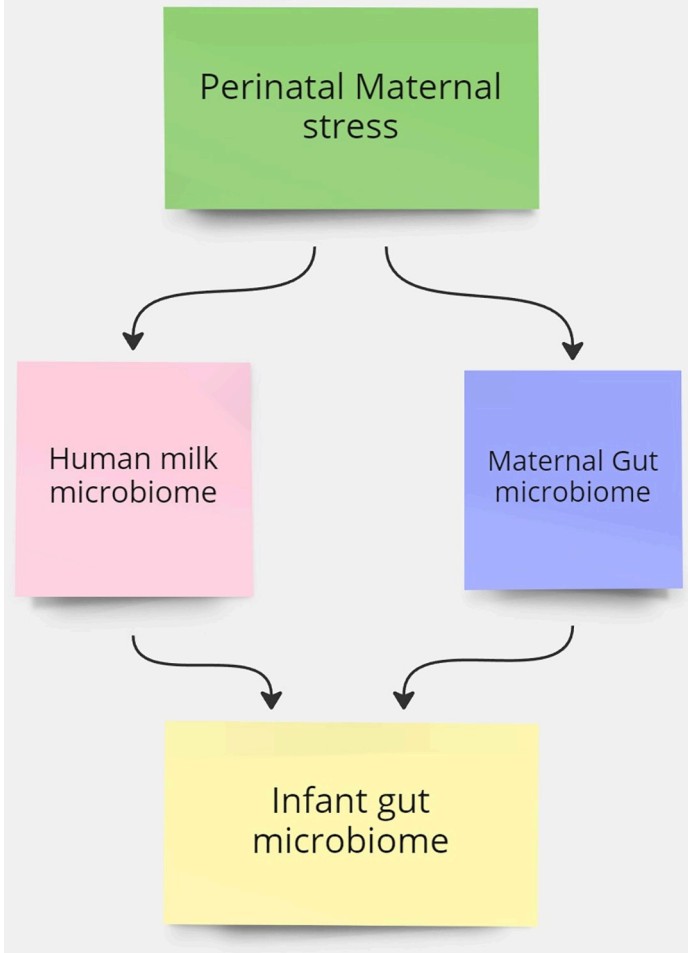

**Fig 1. Concept map.**

and extracting the data, but also in evaluating its credibility due to the absence of standardized quality indicators or peer review processes [53]. The below Table 3 identifies final eligibility criteria for types and sources of evidence to be included in the protocol and full review.

## Search strategy

The review will follow the three-step JBI process: To begin initial searches will be conducted in PubMed and CINAHL to identify relevant papers. Secondly text words, keywords, and index terms will be analysed from identified articles to develop a comprehensive search strategy for all databases (See S1 File). Finally, reference lists of identified articles will be searched for additional studies. Keywords and search terms will be peer-reviewed using PRESS guidance with a librarian. To ensure comprehensive results, there will be multiple searches, one for each concept (See S1 Table). All searches will use Boolean operators AND and OR in title (TI) and abstract searches (AB) with CINAHL Headings and Mesh headings used as appropriate. Information will be sourced from the following databases CINAHL Complete, Psych Info, PubMed, Web of Science and Scopus. The review will restrict analysis to English articles, however initial searches will include all languages to determine the number if any in other languages to avoid bias.

**Table 2. Inclusion and exclusion criteria.**

| | | Inclusion Criteria | Exclusion Criteria |
|---|---|---|---|
| **Population:** | Mothers | • Mothers from conception up to 6 months post-natal | • Mothers who are more than 6 months post- natal. |
| | Infants | • Infants from birth to 6 months old who are exclusively breastfeeding. | Infants older than 6 months. Preterm infants<br>• Infants born with birth defects/anomalies or<br>• (*Infants who combination feed or formula feed. |
| **Concepts:** | Maternal stress | • Studies reported on maternal stress identified by blood sampling or self-assessment tools.<br>• Studies reporting on maternal stress and anxiety will be included if data on stress can be extrapolated as a single variable. | • Studies reporting on other mental health disorders such as depression, anxiety, bipolar, psychosis. |
| | Gut Microbiome | • Studies reporting on the gut microbiome as key outcome of mothers or infants in the above ages with laboratory analysis and sequencing of faecal microbiome. | • Studies focused on neurodevelopmental/ neurocognitive or allergic diseases as key outcomes along with altered gut microbiome.<br>• Studies evaluating probiotic treatments.<br>• Non-Human studies |
| | Human milk microbiome | • Studies focused on human milk microbiome (HMM) of the breastfeeding mother.<br>• Studies including discussion on all composition of breast milk will be included if data can be extracted regarding the human milk microbiome. | • Studies conducted with animals or non-human subjects.<br>• Studies evaluating probiotic treatments.<br>• Studies focused on other components of Human milk with no reference to the Human milk microbiome. |
| **Context** | Perinatal period | • Studies focus on mother from conception to 6 months<br>• Studies focusing on babies up to 6 months old who are exclusively breastfeeding. | • Studies that include maternal or infant data outside this specific perinatal period |

## Source of evidence selection

Following the search, all included studies will be exported to Covidence, and all duplicates removed. Two reviewers (NR HM LP SOM PLW) will independently screen all papers in 2 stages: title and abstract screening, and full-text screening. Agreement from two reviewers will be required for an article to be excluded at the title-screening stage. The full text of selected papers will be assessed against the inclusion criteria by two reviewers. Reasons for exclusion of sources of evidence at full text that do not meet the inclusion criteria will be recorded and reported in the scoping review. Any conflicts in the screening stage will be resolved through the inclusion of a third reviewer. The results of the search will be presented in a Preferred Reporting Items for Systematic Reviews and Meta-Analyses (PRISMA) flow diagram [54] (Fig 2).

**Table 3. Final eligibility criteria.**

| Included | Excluded |
|---|---|
| **Primary Research:**<br>**Experimental and**<br>**Observational Studies:**<br>  • **Quasi- experimental**<br>  • **Cohort studies**<br>  • **Case control**<br>  • **Cross sectional**<br>  • **Case series**<br>**Qualitative data** | |
| **Secondary research:**<br>  • **Systematic reviews and**<br>**meta- analysis** | Secondary Research:<br>• Narrative reviews |
| **Years of Publication 2014 –**<br>**Present** | Before 2014 |
| | Grey literature or unpublished literature including case reports, book chapters, guidelines, commentaries, editorials, letters to the editors. |
| **English** | Non-English |

**PRISMA 2020 flow diagram for new systematic reviews which included searches of databases and registers only**

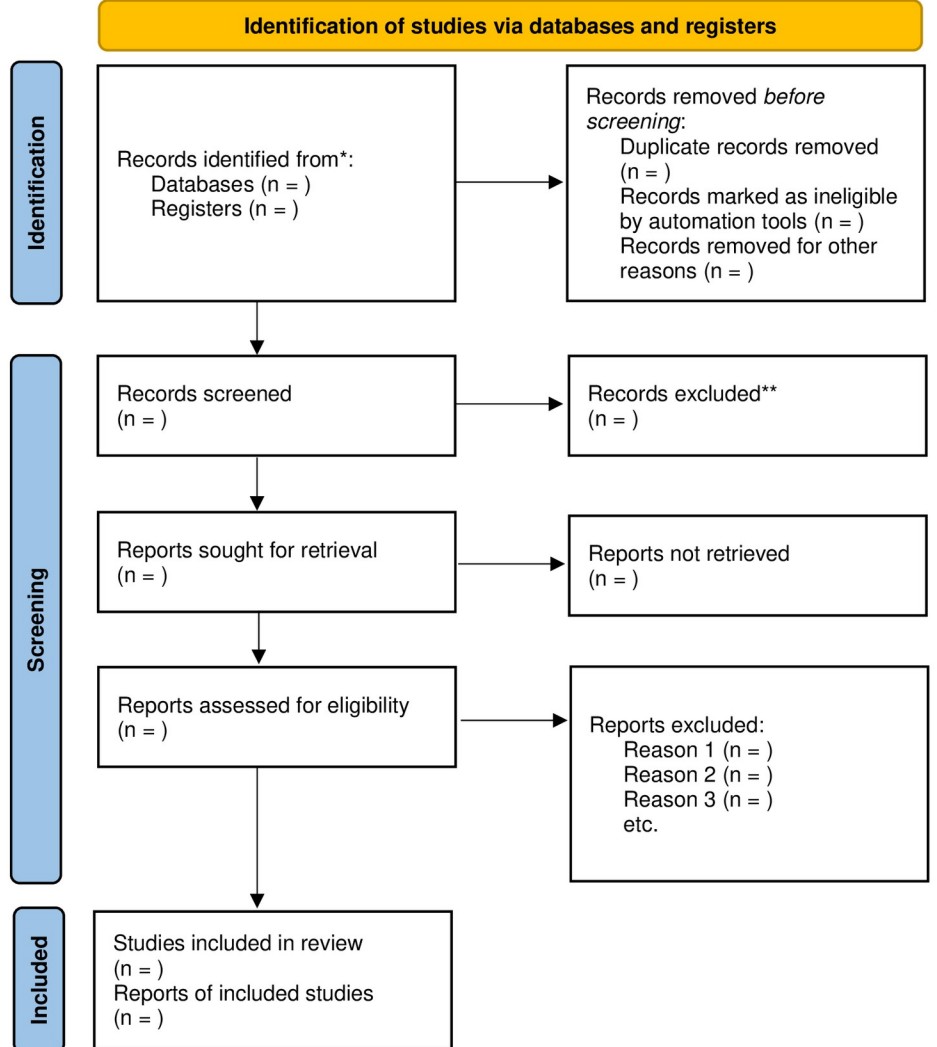

**Fig 2. Prisma flow diagram.** *Consider, if feasible to do so, reporting the number of records identified from each database or register searched (rather than the total number across all databases/registers). **If automation tools were used, indicate how many records were excluded by a human and how many were excluded by automation tools. *From*: Page MJ, McKenzie JE, Bossuyt PM, Boutron I, Hoffmann TC, Mulrow CD, et al. The PRISMA 2020 statement: an updated guideline for reporting systematic reviews. BMJ 2021;372:n71. doi: 10.116/bmj.n71. For more information, visit: http://www.prisma-statement.org/.

## Data extraction

Data will be extracted from papers included in the scoping review by two reviewers using a data extraction tool developed by the reviewer NR on Covidence (S2 File). Data extracted will include author's name, year, and type of publication. The aims/purpose in relation to the concepts and objectives will be extracted along with methodology and methods, sample (e.g. maternal/ infant or both). Analysis of confounding variables will be noted along with statistical tests and key findings that relate to the scoping review question with limitations/quality issues noted. The extraction table will be piloted on two papers to ensure all relevant results are extracted and to improve transparency and consistency [39, 46].

### Data analysis and presentation

The results will be presented in a narrative summary, along with tables and charts. Gaps in the evidence will be identified at this stage. Preferred Reporting Items for Systematic Reviews and Meta-analyses extension for Scoping review (PRISMA-ScR) will be completed along with presentation of results to ensure each element is covered within the scoping review (S3 File).

### Supporting information

**S1 File. Detailed search strategy.**
(DOCX)

**S2 File. Data extraction template for covidence.**
(DOCX)

**S3 File. PRISMA Sr checklist completed.**
(PDF)

**S4 File. Completed PRISMA P checklist.**
(DOCX)

**S1 Table. Keywords in multiple searches.**
(DOCX)

### Acknowledgments

This scoping review was produced by author Niamh Ryan NR as a contribution to her Doctoral degree.

### Author Contributions

**Conceptualization:** Niamh Ryan, Patricia Leahy-Warren, Helen Mulcahy, Siobhain O'Mahony, Lloyd Philpott.

**Data curation:** Niamh Ryan, Patricia Leahy-Warren, Siobhain O'Mahony, Lloyd Philpott.

**Formal analysis:** Niamh Ryan, Patricia Leahy-Warren, Siobhain O'Mahony, Lloyd Philpott.

**Investigation:** Niamh Ryan.

**Methodology:** Niamh Ryan, Patricia Leahy-Warren, Helen Mulcahy, Siobhain O'Mahony, Lloyd Philpott.

**Project administration:** Niamh Ryan, Helen Mulcahy.

**Supervision:** Patricia Leahy-Warren, Helen Mulcahy, Siobhain O'Mahony, Lloyd Philpott.

**Validation:** Niamh Ryan, Patricia Leahy-Warren, Helen Mulcahy, Siobhain O'Mahony, Lloyd Philpott.

**Visualization:** Niamh Ryan, Patricia Leahy-Warren, Helen Mulcahy, Siobhain O'Mahony, Lloyd Philpott.

**Writing – original draft:** Niamh Ryan.

**Writing – review & editing:** Niamh Ryan, Patricia Leahy-Warren, Helen Mulcahy, Siobhain O'Mahony, Lloyd Philpott.

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
