## [Decision Letter · Decision Letter 0]

9 May 2024

PONE-D-24-12413The impact of perinatal maternal stress on the maternal and infant gut and human milk microbiomes: A scoping review protocolPLOS ONE

Dear Dr. Ryan,

Thank you for submitting your manuscript to PLOS ONE. After careful consideration, we feel that it has merit but does not fully meet PLOS ONE’s publication criteria as it currently stands. Therefore, we invite you to submit a revised version of the manuscript that addresses the points raised during the review process.

We look forward to receiving your revised manuscript.

Kind regards,

Wafaa Rashed, PhD

Academic Editor

PLOS ONE

Journal Requirements:

3. We note that you have referenced (unpublished) on page 7, which has currently not yet been accepted for publication. Please remove this from your References and amend this to state in the body of your manuscript: (ie “Bewick et al. [Unpublished]”) as detailed online in our guide for authors

4. Please upload a copy of Figure 1, to which you refer in your text on page 4. If the figure is no longer to be included as part of the submission please remove all reference to it within the text.

5. We notice that your supplementary [S Fig 1 and S Fig 2 ] are included in the manuscript file. Please remove them and upload them with the file type 'Supporting Information'. Please ensure that each Supporting Information file has a legend listed in the manuscript after the references list.

Reviewers' comments:

Reviewer's Responses to Questions

**Comments to the Author**

1. Does the manuscript provide a valid rationale for the proposed study, with clearly identified and justified research questions?

Reviewer #1: Yes

Reviewer #2: Yes

2. Is the protocol technically sound and planned in a manner that will lead to a meaningful outcome and allow testing the stated hypotheses?

Reviewer #1: Yes

Reviewer #2: Partly

3. Is the methodology feasible and described in sufficient detail to allow the work to be replicable?

Reviewer #1: Yes

Reviewer #2: Yes

4. Have the authors described where all data underlying the findings will be made available when the study is complete?

Reviewer #1: Yes

Reviewer #2: Yes

5. Is the manuscript presented in an intelligible fashion and written in standard English?

Reviewer #1: Yes

Reviewer #2: Yes

6. Review Comments to the Author

You may also provide optional suggestions and comments to authors that they might find helpful in planning their study.

Reviewer #1: The authors address an important public health topic. The protocol is detailed and well-written.

Minor comments to consider

- Kindly correct the numbering of "The S File 2. Data extraction template for Covidence". Upon download, it shows as S File 3. Data extraction template for Covidence

- For the data extraction sheet, you might consider specifying the biomarker tested for studies in which blood sampling was carried out and separately reporting Shannon α, β as measures of diversity.

Reviewer #2: Ryan et al. designed a scoping review protocol aiming to analyse the impact of perinatal maternal stress on the maternal and infant gut and human milk microbiomes.

I have few comments:

1-In view of the limited number of studies that focus on the effect of maternal perinatal stress on gut microbiome, I recommend that the author may consider including data sources in addition to the peer reviewed articles and systematic reviews, such as book chapters and guideline. Although the authors mentioned the challenges they faced during data extraction from those data sources, but looking into these sources may avoid missing potentially relevant data.

2-In order to improve the search results, the authors may consider adding some key words such as metagenomics, biofilms, dysbiosis and microflora.

3-Finally, I recommend adding a few sentences to the manuscript discussing how would this scoping review help in improving the health of pregnant women and newborn.

7. PLOS authors have the option to publish the peer review history of their article (what does this mean?). If published, this will include your full peer review and any attached files.

Reviewer #1: No

Reviewer #2: No

---

## [Author Response · Author response to Decision Letter 0]

16 May 2024

Thank you to reviewer 1 and 2 for concise and helpful direction. I have responded to all comments in response to reviewer letter.

---

## [Editor Report · Decision Letter 1]

20 May 2024

The impact of perinatal maternal stress on the maternal and infant gut and human milk microbiomes: A scoping review protocol

PONE-D-24-12413R1

Dear Dr. Ryan,

We’re pleased to inform you that your manuscript has been judged scientifically suitable for publication and will be formally accepted for publication once it meets all outstanding technical requirements.

Kind regards,

Wafaa Rashed, PhD

Academic Editor

PLOS ONE
---

## [Editor Report · Acceptance letter]

24 May 2024

PONE-D-24-12413R1 

PLOS ONE

Dear Dr. Ryan, 

I'm pleased to inform you that your manuscript has been deemed suitable for publication in PLOS ONE. Congratulations! Your manuscript is now being handed over to our production team.

Kind regards, 

on behalf of

Dr. Wafaa Rashed 

Academic Editor

PLOS ONE